**Data Availability Statement:** Data contain potentially identifying or sensitive patient information / study location specific information

# Unsterile injection equipment associated with HIV outbreak and an extremely high prevalence of HCV—A case-control investigation from Unnao, India

Sandip Patil[1☉], Amrita Rao[1☉], Preety Pathak[2], Swarali Kurle[1], Arati Mane[1], Amit Nirmalkar[1], A. K. Singhal[3], Vinita Verma[4], Mukesh Kumar Singh[3], D. C. S. Reddy[5], Ashwini Shete🆔[1], Manjula Singh[6], Raman Gangakhedkar[6], Samiran Panda🆔[1,6]*

1 Indian Council of Medical Research-National AIDS Research Institute, Pune, Maharashtra, India, 2 Uttar Pradesh State AIDS Control Society, Lucknow, Uttar Pradesh, India, 3 Community Health Centre, Department of Medical & Health, Government of Uttar Pradesh, Bangarmau, India, 4 National AIDS Control Organization, New Delhi, India, 5 Technical Resource Group, National AIDS Control Organization, New Delhi, India, 6 Indian Council of Medical Research Headquarter, New Delhi, India

☉ These authors contributed equally to this work.
* director@nariindia.org, pandasamiran@gmail.com, samiranpanda.hq@icmr.gov.in

## Abstract

The integrated counseling and testing center (ICTC) located in the district hospital, Unnao in the northern state of Uttar Pradesh (UP), India witnessed an increased detection of HIV among its attendees in July 2017. Subsequently, health camps were organized by the UP State AIDS Control Society in the villages and townships contributing to such detection. We conducted a case-control study to identify factors associated with this increased detection; 33 cases and 125 controls were enrolled. Cases were individuals, detected HIV sero-reactive during November 2017-April 2018 from three locations namely Premganj, Karimuddin-pur and Chakmeerapur in the Bangarmau block of the district of Unnao. Controls hailed from the same geographical setting and tested HIV sero-nonreactive either in health camps or at ICTC centers from where the cases were detected. Misclassification bias was avoided by confirming HIV sero-status of both cases as well as controls prior to final analysis. Study participants were interviewed on various risk practices and invasive treatment procedures. They were also tested for HIV and other bio-markers reflecting unsafe injecting and sexual exposures such as hepatitis B surface antigen (HBsAg), anti-HCV antibody (HCV Ab), anti-herpes simplex-2 Immunoglobulin G (HSV-2 IgG) and rapid plasma regain (RPR) test for syphilis. Secondary data analysis on three time points during 2015 through 2018 revealed a rising trend of HIV among attendees of the ICTCs (ICTC-Hasanganj, ICTC-Unnao district hospital and ICTC- Nawabganj) catering to the entire district of Unnao. While there was a seven fold rise of HIV among ICTC attendees of Hasanganj ($\chi^2$ value for trend 23.83; $p < 0.001$), the rise in Unnao district hospital was twofold ($\chi^2$ value for trend 4.37; $p < 0.05$) and was tenfold at ICTC-Nawabganj ($\chi^2$ value for trend 5.23; $p < 0.05$) indicating risk of infection prevailing throughout the district. Primary data was generated through interviews and laboratory investigations as mentioned above. The median age of cases and controls was 50

which, if disclosed, may lead to stigma and discrimination. Due to ethical restrictions, data can be made available on request to Institutional Ethics committee of ICMR-NARI, Pune. (Email id-ecnari@nariindia.org) as well as Project Director, Uttar Pradesh State AIDS Control Society (UPSACS) (E mail id- pd.upsacs@gmail.com).

**Funding:** This case control study received funds from Indian Council of Medical Research, New Delhi under Short Term New Scheme (Order no. HIV/50/183/2/2018-ECD II, dated 22-03-2018). The funders had no role in study design, data collection and analysis, decision to publish, or preparation of the manuscript.

**Competing interests:** The authors have declared that no competing interests exist.

year (minimum 18 –maximum 68; IQR 31–57) and 38 year (minimum 18 –maximum 78; IQR 29–50) respectively. Thirty six percent of the cases and 47% of controls were male. A significantly higher proportion of cases (85%) had HCV Ab compared to controls (56%; OR 4.4, 95% CI 1.5–12.1); none reported injection drug use. However, cases and controls did not differ significantly regarding presence of HSV-2 IgG (6% versus 8% respectively). Neither any significant difference existed between cases and controls in terms of receiving blood transfusion, undergoing invasive surgical procedures, tattooing, tonsuring of head or skin piercing. In multivariate logistic regression model, 'unsafe injection exposure during treatment-seeking'(AOR 6.61, 95% CI 1.80–24.18) and 'receipt of intramuscular injection in last five years' (AOR 7.20, 95% CI 1.48–34.88) were independently associated with HIV sero-reactive status. The monophyletic clustering of HIV sequences from 14 cases (HIV-1 *pol* gene amplified) indicated a common ancestry. Availability of auto-disabled syringes and needles, empowerment of the local communities and effective regulatory practices across care settings would serve as important intervention measures in this context.

## 1. Introduction

1986 marks the year when HIV was detected first in India. A group of female sex workers (FSWs) in the southern state of Tamil Nadu was detected by HIV sero-reactive (then termed HTLV-III) [1]. The eastern metropolitan city of Kolkata in West Bengal [2, 3] also witnessed a similar phenomenon in the same year. Close to the heels of these events, a rapid rise of HIV was recorded among FSWs in Mumbai [4] in the western state of Maharashtra. Three years later, Manipur, Mizoram and Nagaland, the north-eastern states of the country bordering Myanmar, witnessed an explosive spread of HIV among people who inject drugs (PWID) [5, 6].

The aforementioned investigations helped in characterizing the concentrated nature of the HIV epidemic in most at-risk population groups (MARPs) in India and its generalized spread in a few States and also guided intervention responses. Strategic planning and operational guidelines developed by the National AIDS Control Organization (NACO) subsequently helped in the containment of the further spread of the virus [7]. However, two decades later, HIV transmission in newly identified pockets of injection drug use in the northern states of Punjab, Uttarakhand, Haryana and Uttar Pradesh (UP) as well as in the states of Odisha, Bihar, Tripura, Karnataka and Arunachal Pradesh raised an alarm [8, 9]. A striking turn of event in July 2017 added to the complexity. HIV was detected in an increasing number among attendees of the Integrated Counseling and Testing Centre (ICTC) located in the district hospital of Unnao, UP. Consequently, the medical superintendent of the hospital brought it to the notice of the district Chief Medical Officer of Health (CMOH) and sought advice. Uttar Pradesh State AIDS Control Society (UPSACS) in response to such an alert, organized health camps in the Premganj township and villages of Karimuddinpur and Chakmeerapur in Bangarmau block (blocks are the administrative subdivisions of a district in India). The first camp was held in November 2017 and an HIV test facility was offered along with health examination to the camp attendees. The decision to organize health camps in the above-specified locations was guided by the contribution of villages and township to HIV detection at the ICTC— Unnao district hospital.

The current case-control investigation was initiated against this backdrop. The overall purpose of our study was to identify factors associated with spiked detection of HIV in the above geographical settings and to suggest appropriate intervention measures.

## 2. Methodology

### 2.1 Study area

Uttar Pradesh (UP), one of the larger states of India, with 75 districts, has a population of 199,581,477 according to the census 2011 [10]. Unnao–the study district in the state—has a land area of 4,558 sq. km with a population size of 3,108,367 (population density 682 per sq. km) and literacy rates among males and females being 75% and 57% respectively. Of the total 11, 24,744 workers in the district, 40% are cultivators and 30% are agricultural laborers underlining the fact that the society is mostly agrarian [11]. Bangarmau is one of the 16 administrative blocks (subdivision) of the Unnao district catering to a population of 221563. The population of Premganj township, Karimuddinpur and Chakmeerapur villages from Bangarmau block, as per census 2011, were 1216, 728, and 630 respectively.

### 2.2 Study design and period

A case-control study was conducted from September through December 2018. To facilitate enrollment of study participants, community sensitization meetings and consultation with the local health authorities, village administration and state officials were held at different time points.

### 2.3 Ethical clearance

Approval for the current investigation was obtained from the Ethics Committee of the Indian Council of Medical Research–National AIDS Research Institute (ICMR-NARI). Written informed consent was obtained from each of the participants before interviewing them. The identity of the respondents was anonymized by ascribing unique code to each of the interview schedules and linked clinical specimens.

### 2.4 Study population

**2.4.1 Definition—Case.** Cases were individuals detected HIV sero-reactive during the six-month-period (November 2017 to April 2018) from three study locations namely Premganj, Karimuddinpur and Chakmeerapur. Participants ≥18 year (adults) were recruited. Anti-retroviral treatment (ART) centre-Kanpur, ICTC- Hasanganj and ICTC-Unnao district hospital catered to the three study locations and records maintained by them were used to prepare the case-list. Information collected during health camps organized by UPSACS was also utilized to finalize this list.

**2.4.2 Definition—Control.** Controls were individuals, who lived in any of the three locations as with cases and tested HIV sero non-reactive in health camps or at ICTC centers from where cases were detected during the defined period.

**2.4.3 Recruitment of participants.** Five of the 56 cases were minors and not enrolled. Among the remaining 51 adults, 25 were males. Thirty-one cases were available for enrollment and interview. Reasons for non-participation among the rest (20/51) were deaths, non-traceability in community and refusal. Despite our plan to recruit four controls per case, refusal and difficulty in blood collection allowed the final enrollment of 127 controls. To avoid misclassification bias, we re-tested all the enrolled cases and controls for HIV. In the process, two

controls were detected as HIV sero-reactive. Thus the following analysis had 33 cases and 125 controls (Fig 1).

## 3. Study tool and procedures

Being informed by two patients from the Premganj township of Bangarmau block, the medical superintendent of Unnao district hospital wrote to the CMOH that a local treatment provider had allegedly been using the same syringe and needle on different individuals for treatment. Among other things, the study questionnaire incorporated this issue as well. The district health authorities and officials from UPSACS helped in refining this study tool with the inclusion of local terminology for words such as 'tattoo', 'pus' and 'system of medicine'. Positive HIV sero-reactive status was the dependent variable. Socio-demographic profile, substance use practices, injection exposure in therapeutic settings, sexual practices, symptoms of sexually transmitted diseases (STDs), blood transfusion, surgical interventions, dental procedure, barber service utilization, tattooing, tonsuring, skin piercing practices and illness histories were inquired upon as independent variables.

Male and female interviewers, identified from a nearby locality, were trained on the structure and content of the finalized interview schedule through mock and supervised interview sessions. Male and female participants were interviewed by male and female interviewers respectively.

### 3.1 Serologic investigations

Seven milliliters of blood was collected from each study participant in an Ethylene Diamine Tetra Acetic acid (EDTA) vacutainer tube. Plasma separation was carried out by trained project staff at ART-centre, Kanpur and Community Health Centre (CHC) Bangarmau, which served as interview sites as well. Specimens were stored at -18˚C till transportation in dry ice to ICMR-NARI, Pune.

HIV antibody was detected by using a rapid diagnostic method (Meriscreen HIV1-2 and Signal HIV) as well as the Enzyme-Linked Immunosorbent Assay (ELISA) (Microlisa HIV). Antibody against hepatitis C virus (anti-HCV IgG) was tested using Ortho HCV 3.0 ELISA (Ortho Clinical Diagnostics, USA). All HCV sero-reactive specimens were further tested by the Xpert HCV viral load assay (Cepheid, USA). Hepatitis B surface antigen was detected using Murex HBsAg Version 3.0 ELISA (Diasorin, Italy). Rapid Plasma Reagin antibody kit (Arkray Healthcare Pvt Ltd, India) and HSV-2 IgG ELISA test system (DIAPRO, Italy) were used for detecting syphilis and exposure to herpes virus type 2 infection respectively.

### 3.2 HIV-1 pol gene sequencing

Viral RNA extraction was done from the plasma using the NucliSens EasyMag total nucleic acid extraction system (Biomerieux). The extracted RNA was amplified for complete HIV-1 protease and partial reverse transcriptase region as described previously [12] and sequenced using Genetic Analyzer (Applied Biosystems 3130XL, Thermo Fisher Scientific). Sequence assembly and base calling was performed using SeqScape v2.6 software (Thermo Fisher Scientific). Sequence FASTA files generated were employed for further analysis.

### 3.3 Data analyses

Analysis of the trend of HIV case detection in all three ICTCs (Hasanganj, Unnao district hospital, and Nawabganj) and examination of primary data generated through one-on-one interviews and blood tests were two principal investigation approaches. Paper-based forms were

## STUDY DISTRICT – UNNAO, UTTAR PRADESH, INDIA

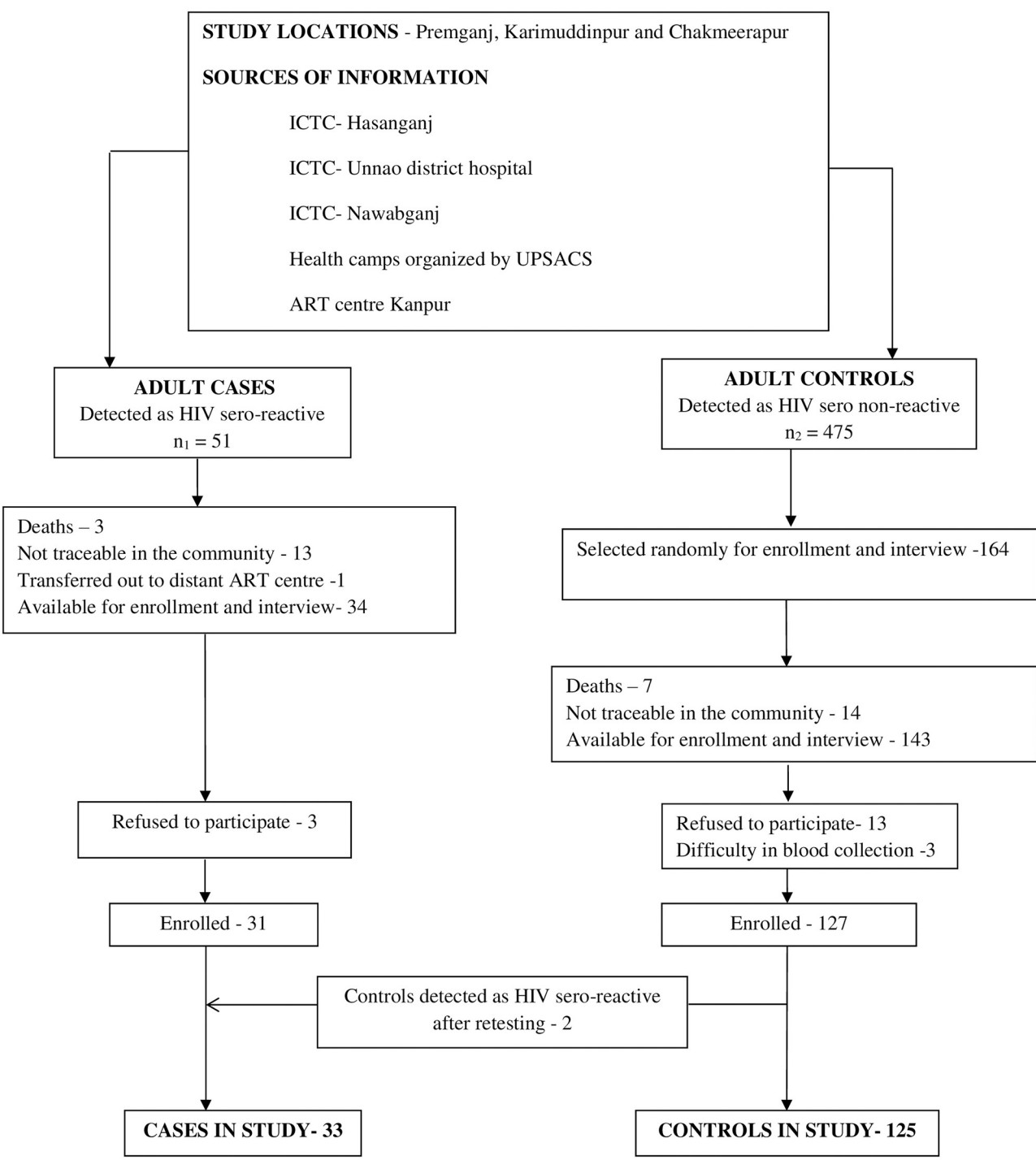

**Fig 1. Flow chart—Recruitment of study population.**

used to capture interview responses, which were checked for their quality daily and computerized following necessary corrections. Distributions of cases and controls across various exposures were compared. Association between exposure variables and HIV sero-reactive status were examined through the Mantel-Haenszel estimate of the odds ratio. Biologically plausible variables and variables bearing significant association (p < 0.1) with the study outcome were entered into a logistic regression model using STATA (version 10.0/10.1; StataCorp, College Station, TX).

## 4. Results

### 4.1 Secondary data analysis

ICTC data generated by UPSACS during 2015–2018 were used. Significantly rising trends of positive HIV sero-reactive status among attendees of all three ICTC centers (Hasanganj, Unnao district hospital and Nawabganj) were evident (Table 1).

### 4.2 Participants' profile

The median age of the cases was 50 year and mean 46 year (minimum 18; maximum 68, IQR 31–57 year), and that of the controls was 38 year and 40 year respectively (minimum 18; maximum 78; IQR 29–50 year). About a third of the cases (12/33) and 47% of controls (59/125) were males. Nearly half of the cases and controls never attended a school.

   None of the cases and controls during study participation reported 'being away from home for work'. About a fifth of the cases (6/33) reported ever staying away from home for 3 months or more. Controls (26/125; 21%) did not differ significantly with cases in this regard. While more than half of the cases reported being unemployed, about a third of the controls reported so (OR 2.38; 95% CI 0.98–5.79; p = 0.055, Table 2). The majority of the males in cases made a living either through farming or as a daily wage laborer (10/12; 83%). Similar was the profile of occupational engagement among controls (39/59; 66%). Most of the female participants among cases (17/21; 81%) were housewives whereas a lesser proportion in controls reported so (42/64; 66%).

### 4.3 Sexual practices and STD symptoms

One-tenth of the cases (3/33) and controls (12/125) were unmarried. The median age at onset of sexual intercourse for males in cases was 19 year (minimum 17; maximum 45, IQR 18–33 year) and among females was 16 year (minimum 13; maximum 20, IQR 14–17 year). The median age at onset of sexual intercourse in controls among males was 19 year (minimum 12; maximum 35, IQR 17–23 year) and that in females was 18 year (minimum 14; maximum 21, IQR 16–19 year). Data on sexual practices and injection exposure during treatment seeking are presented in Table 2. No significant difference was observed between cases and controls on

**Table 1. HIV seroreactivity among ICTC attendees, Unnao district.**

| ICTC centers, Unnao district | HIV test | 2015–16 | 2016–17 | 2017–18 | Chi-square value (trend) | p-value |
|---|---|---|---|---|---|---|
| CHC Hasanganj | Positive | 3 | 10 | 42 | 23.83 | < 0.001 |
|  | Negative | 1287 | 1501 | 2065 |  |  |
| District hospital, Unnao | Positive | 53 | 47 | 82 | 4.37 | 0.03 |
|  | Negative | 6528 | 6033 | 7058 |  |  |
| CHC Nawabganj | Positive | 1 | 4 | 6 | 5.23 | 0.02 |
|  | Negative | 1431 | 1515 | 948 |  |  |

**Table 2. Socio-demographic profile, sexual practices* and injection exposure.**

| Practices | Cases | Controls | OR[a] (95% CI[b]) | p-value |
|---|---|---|---|---|
| | n (%) | n (%) | | |
| **Age** | | | | |
| > 37 year | 23 (69.7) | 69 (55.2) | 1.86 (0.82–4.25) | 0.137 |
| ≤ 37 year | 10 (30.3) | 56 (44.8) | Reference | |
| **Residence** | | | | |
| Premganj | 18 (54.5) | 62 (49.6) | 0.48 (0.10–2.22) | 0.351 |
| Chakmeerapur | 12 (36.4) | 58 (46.4) | 0.34 (0.07–1.64) | 0.181 |
| Karimuddinpur | 3 (9.1) | 5 (4) | Reference | |
| **Occupation** | | | | |
| Unemployed | 19 (57.58) | 46 (36.80) | 2.38 (0.98–5.79) | 0.055 |
| Famer | 5 (15.15) | 27 (21.60) | 1.06 (0.33–3.50) | 0.911 |
| Non-agricultural | 9 (27.27) | 52 (41.60) | Reference | |
| **Ever had sex with a female casual partner as reported by male participants** | | | | |
| Yes | 2 (20.0) | 4 (6.9) | 3.38 (0.53–21.52) | 0.19 |
| No | 8 (80.0) | 54 (93.1) | Reference | |
| **Condom use during last sex** | | | | |
| No | 24 (82.8) | 91 (80.5) | 1.16 (0.39–3.38) | 0.785 |
| Yes | 05 (17.2) | 22 (19.5) | Reference | |
| **Received intramuscular injection in last five years** | | | | |
| Yes | 31 (94.0) | 79 (63.2) | 9.02 (2.06–39.46) | 0.003 |
| No | 02 (6.0) | 46 (36.8) | Reference | |
| **Syringe & needle during intramuscular injection in the last five years** | | | | |
| Didn't notice if the syringe & needle were new | 6 (18.18) | 11 (8.8) | 3.67 (1.18–11.32) | 0.024 |
| Injected by used syringe & needle | 9 (27.27) | 5 (4.0) | 11 (3.30–36.57) | < 0.001 |
| Injected by new syringe & needle | 18 (54.55) | 109 (87.2) | Reference | |

[a] Odds Ratio;

[b] Confidence Interval

*None among male participants from cases and only two participants from controls reported ever having sex with female sex workers.

self-reported STD symptoms experienced over the last year except for anal ulcers (7% in cases compared to 1% in controls). Overall, very few cases and controls (< 3%) reportedly experienced genito-ulcerative or genito-secretory disease symptoms.

## 4.4 Sexually transmitted infection biomarkers and substance use

None of the participants was sero-reactive for syphilis. Table 3 presents the distribution of cases and controls across HBsAg and HSV-2 IgG without significant difference between groups. Two cases and one control could not be tested for HSV-2 IgG as serum specimens were inadequate. A striking difference in exposure to hepatitis C (indicated by the presence of HCV Ab) was identified between cases (28/33; 85%) and controls (70/125; 56%). Eighty-seven percent of people living with HIV (PLHIV cases), and 72% of controls, who were HCV sero-reactive showed the presence of HCV RNA.

About 40% of men in cases and a similar proportion in controls reported ever having a drink containing alcohol. The finding was corroborated while women participants were asked about alcohol use by their spouses. None of the participants reported ever injecting drugs for non-medicinal or recreational purposes.

Table 3. Unsafe injecting and sexual exposure-related biomarkers.

| Characteristics | Cases | Controls | OR[a] (95% CI[b]) | p-value |
|---|---|---|---|---|
| | n (%) | n (%) | | |
| **HCV antibody** | | | | |
| Sero-reactive | 28 (84.8) | 70 (56) | 4.4 (1.5–12.1) | 0.004 |
| Sero-nonreactive | 5 (15.2) | 55 (44) | Reference | |
| **HBsAg** | | | | |
| Sero-reactive | 2 (6.1) | 5 (4.0) | 1.5 (0.28–8.36) | 0.611 |
| Sero-nonreactive | 31 (93.9) | 120 (96) | Reference | |
| **HSV-2 IgG** | | | | |
| Sero-reactive | 2 (6.5) | 10 (8.1) | 0.79 (0.16–3.78) | 0.764 |
| Sero-nonreactive | 29 (93.5) | 114 (91.9) | Reference | |

[a] Odds Ratio;

[b] Confidence Interval.

### 4.5 Health seeking and HCV exposure

A significantly greater proportion of cases (32/33; 97%) sought professional help for fever, body ache, common cold and cough experienced in the last one year compared to controls; 69/125; 55% (OR 25.97; 95% CI 3.44–196; p = 0.002); plausibly indicating more illness experiences in them. We also compared cases (PLHIV) against controls regarding receipt of intravenous fluid and intramuscular injection during treatment-seeking within the last five years (Table 2), which turned out to be significantly different (97% vs 64% and 94% vs 63% respectively). This difference could be explained by the greater proneness of PLHIV to fall sick compared to controls. To investigate unsafe injection exposure, we sought treatment history over the last five years with specific attention to the multi-use of syringe and needle. A significantly higher proportion of cases reported being exposed to used syringe and needle compared to controls while seeking treatment (27% and 4% respectively; OR 11; 95% CI 3.3–36.5; p = < 0.001; Table 2). The bar diagram in Fig 2 depicts exposure to HCV infection across all age groups in cases and controls indicating a concerning level of transmission in the study community.

### 4.6 Blood transfusion and invasive procedures

Cases and controls did not differ significantly in experiencing 'surgical procedures', 'blood transfusion', 'dental procedures', 'tattooing', 'skin-piercing' and 'tonsuring of the head' (Table 4).

### 4.7 Multivariate analysis

In the multivariate model, we included residential location as well as age as two attributes because they could reflect various confounders about which information could not be collected. Five other variables from univariate analyses, due to their statistical significance (< 0.1), and biologic plausibility qualified to be included in the multivariate logistic model. These were occupation, receipt of intramuscular injection in the last five years, exposure to used syringe and needle while seeking treatment, HCV Ab sero-reactive status and anal ulcer. We did not include HCV Ab as one of the explanatory variables in the multivariate model, because unsafe injection exposure rather than HCV would serve as a biologically plausible factor for HIV acquisition. We also did not include an anal ulcer as this was not a clinically

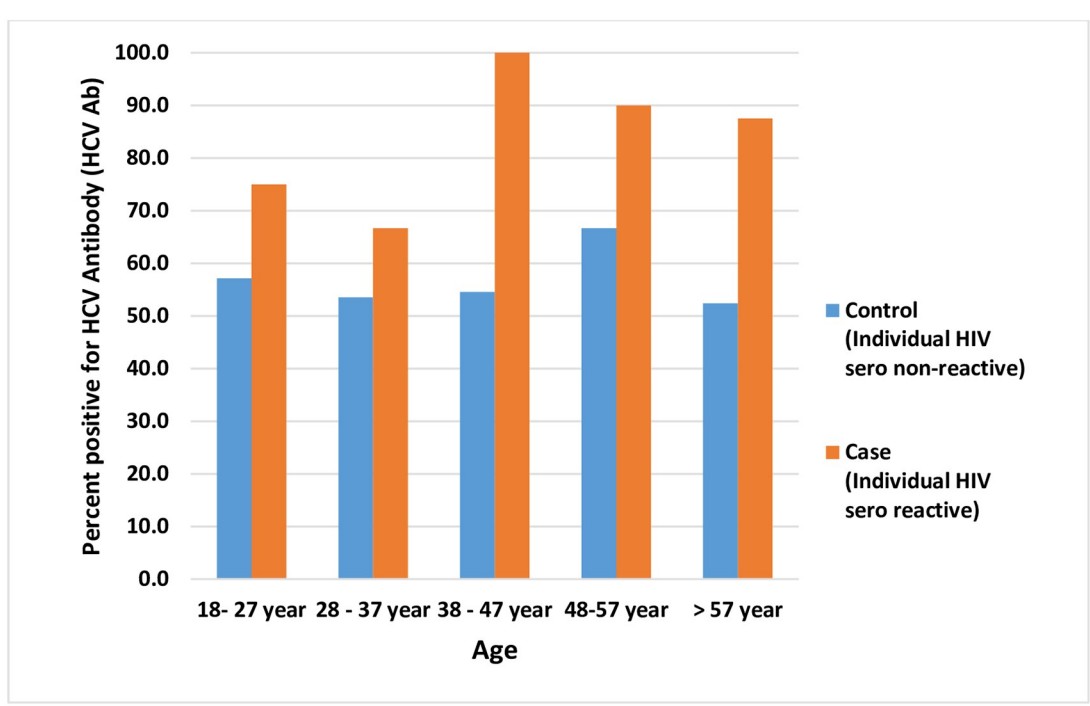

**Fig 2. HCV sero-reactive status in HIV infected and non-infected individuals.**

Table 4. Blood transfusion and invasive procedures.

| Invasive procedure in the last 5 years | Case n (%) | Control n (%) | OR[a] (95% CI[b]) | p-value |
|---|---|---|---|---|
| **Surgical procedure** | | | | |
| Yes | 3 (9.1) | 14 (11.2) | 0.79 (0.21–2.94) | 0.728 |
| No | 30 (90.9) | 111 (88.8) | Reference | |
| **Blood transfusion** | | | | |
| Yes | 1 (3.1) | 2 (1.6) | 1.98 (0.17–22.59) | 0.581 |
| No | 32 (96.9) | 123 (98.4) | Reference | |
| **Dental procedure** | | | | |
| Yes | 8 (24.24) | 39 (31.2) | 0.71 (0.29–1.70) | 0.438 |
| No | 25 (75.76) | 86 (68.8) | Reference | |
| **Tattooing** | | | | |
| Yes | 4 (12.1) | 6 (4.8) | 2.74 (0.72–10.32) | 0.138 |
| No | 29 (87.8) | 119 (95.2) | Reference | |
| **Skin piercing** | | | | |
| Yes | 1 (3) | 4 (3.2) | 0.94 (0.1–8.75) | 0.961 |
| No | 32 (96.9) | 121 (96.8) | Reference | |
| **Tonsuring** | | | | |
| Yes | 10 (30.3) | 48 (39) | 0.67 (0.29–1.55) | 0.359 |
| No | 23 (69.7) | 75 (60.9) | Reference | |

[a] Odds Ratio;

[b] Confidence Interval.

**Table 5. Factors associated with HIV infection in multivariate analysis.**

| Variable | Cases | Controls | AOR[a] (95% CI[b]) | p-value |
|---|---|---|---|---|
| | n (%)[a] | n (%) | | |
| **Age** | | | | |
| ≤ 37 year | 10 (30.3) | 56 (44.8) | Reference | |
| > 37 year | 23 (69.7) | 69 (55.2) | 2.07 (0.79–5.37) | 0.134 |
| **Residence** | | | | |
| Premganj | 18 (54.5) | 62 (49.6) | 0.28 (0.5–1.62) | 0.157 |
| Chakmeerapur | 12 (36.4) | 58 (46.4) | 0.14 (0.02–0.9) | 0.039 |
| Karimuddinpur | 3 (9.1) | 5 (4) | Reference | |
| **Occupation** | | | | |
| Unemployed | 19 (57.58) | 46 (36.80) | 2.10 (0.77–5.72) | 0.143 |
| Farmer | 5 (15.15) | 27 (21.60) | 1.06 (0.25–4.48) | 0.936 |
| Non-agricultural | 9 (27.27) | 52 (41.60) | Reference | |
| **Received intramuscular injection in last five years** | | | | |
| Yes | 31 (94.0) | 79 (63.2) | 7.20 (1.48–34.88) | 0.014 |
| No | 02 (6.0) | 46 (36.8) | Reference | |
| **Syringe & needle used while receiving intramuscular injection during the last five years** | | | | |
| Didn't notice | 6 (18.18) | 11 (8.8) | 2.81(0.81–9.69) | 0.1 |
| Injected by used syringe & needle | 9 (27.27) | 5 (4.0) | 6.61 (1.80–24.18) | 0.004 |
| Injected by new syringe & needle | 18 (54.55) | 109 (87.2) | Reference | |

[a] Adjusted Odds Ratio;

[b] Confidence Interval.

established diagnosis. Adjusting for age, residential location and the rest of the three variables, the following two factors were found to be independently associated with HIV positive serostatus, a) 'getting injected by a used syringe and needle', and b) 'receipt of intramuscular injection in last five years' (Table 5). Unemployment, in both univariate and multivariate analyses, drew our attention towards an indicative association.

## 4.9 Phylogenetic analysis

The pol gene sequences spanning HXB2 coordinates 2253–3281 from study participants (n = 14) were combined with HIV-1 subtype reference sequence alignment obtained from Los Alamos HIV sequence database (https://www.hiv.lanl.gov/content/sequence/NEWALIGN/align.html). These sequences were aligned with the help of MEGA 6 [13] and the alignment was subjected to phylogenetic analysis. Briefly, a maximum likelihood tree was constructed with the help of IQ-Tree version 1.2 [14]. The phylogenetic tree was constructed using a general time-reversible substitution model with a gamma-distributed rate of variation and a proportion of invariant sites (GTR+G+I) with 1000 bootstrap replicates. It was evident from this analysis that most of the *pol* sequences were matching best with Indian HIV-1 C subtype sequences from India. All sequences were monophyletic and had a common ancestor (Fig 3).

We identified factors associated with HIV outbreak in select locations of Bangarmau block of the study district. The way of living in these locations is mostly agrarian. Moreover, we detected a concomitant spread of HCV in the study population; 85% of the PLHIV and 56% of the controls (HIV sero non-reactive) had evidence of exposure to hepatitis C infection. In addition to these findings, secondary data analysis during the present investigation revealed a

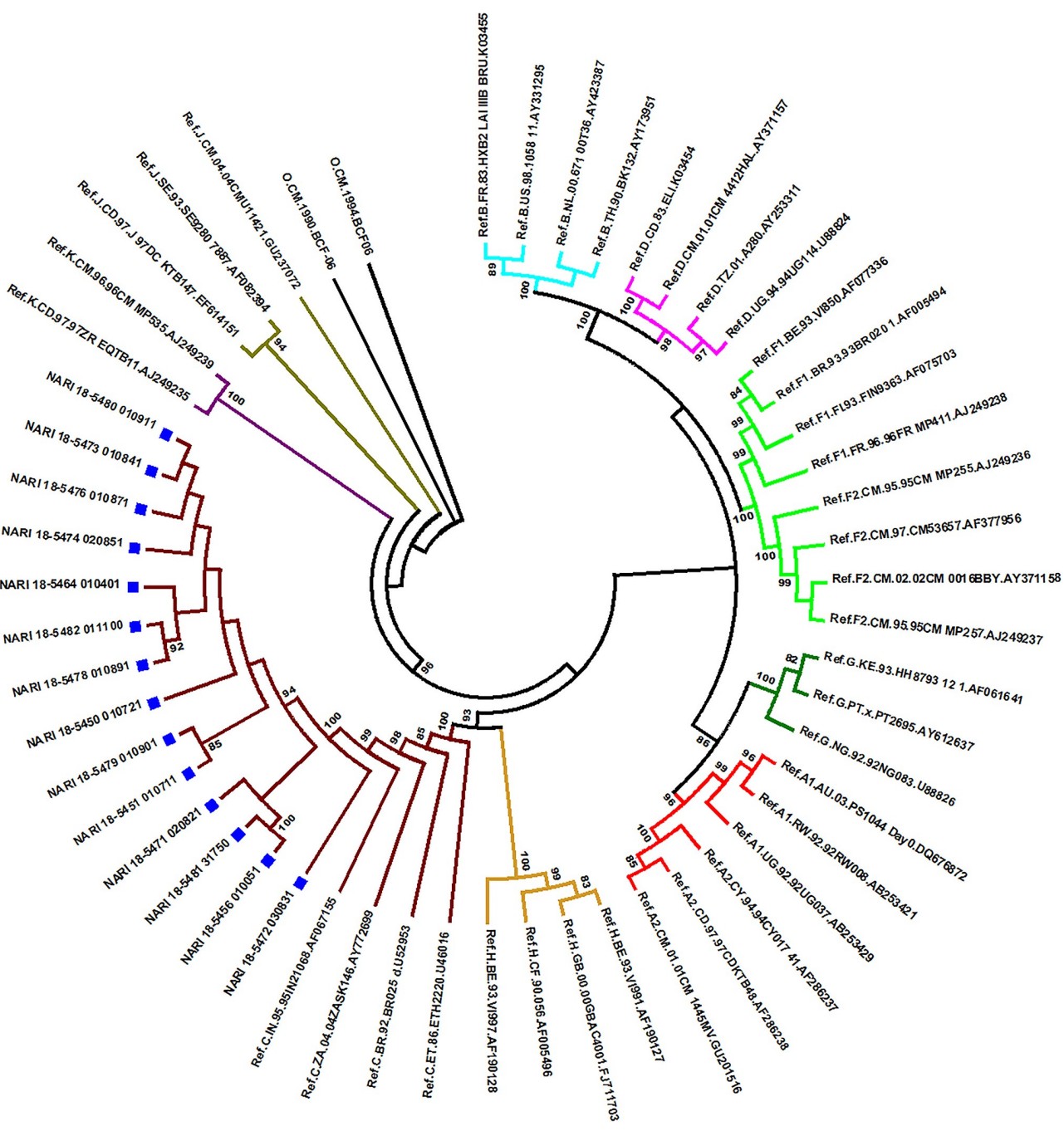

**Fig 3. Maximum likelihood tree showing clustering of HIV-1 subtype C.**

significantly rising trend of HIV among ICTC attendees in the district of Unnao during 2015–2018.

Countries in Asia have recorded HIV transmission due to sharing of injection equipments during drug use as well as faulty injection practices in therapeutic settings. For example, the Larkana district in the northwestern Province of Sindh in Pakistan witnessed the first outbreak of HIV in 2003 among PWID in whom HIV rose from 0.5% to 10% [15]. In 2016, the city of

Larkana experienced spread of HIV among dialysis patients [16]. In the latter instance, information shared by a patient about his own HIV status prompted subsequent investigations.

Noticeably, between the aforementioned two events of HIV detection in Larkana, the district of Gujrat in the Punjab province of Pakistan identified spread of HIV through community based health camps [17]. Later in 2018, another HIV outbreak was reported from a village near Kot Momin, Sargodha situated in Central Punjab of Pakistan, where 1.29% of the inhabitants of the village were found to be HIV infected. Use of contaminated injection equipment by a quack doctor was implicated for this outbreak. Six months later, a survey in general population in the same village recorded 13% HIV prevalence [18]. In the recent past, a concerning level of spread of HIV was noticed during a screening campaign in Larkana. Of the 700 HIV positive cases detected among 26000 screened, 600 were under five children. The World Health Organization (WHO) declared this event as grade II emergency. Unsafe injection during treatment, contaminated blood transfusion and male circumcision with unsterile equipment were found to be the drivers of this spread [19].

Despite lack of injection drug use, presence of HCV Ab in high proportion among study participants in the current investigation was striking. Eighty Five percent of the PLHIV in this study had presence of HCV Ab, which was higher than that in controls (56%). This could be explained by greater illness experiences of PLHIV and consequent exposure to unsafe injecting during treatment seeking. Exposure to HCV among controls (56%) was also of concern and further indicated unsafe injecting in therapeutic setting (Table 3) in the local community. Worth noting here is that HCV rarely spreads through sex [20, 21] and a systematic review has documented HCV prevalence in general population in India to be ranging from 0.44% to 0.88% [22].

The finding that a significantly higher proportion of cases (27%) reported being injected by a syringe and needle already used on others compared to controls (4%), and the aforementioned evidence around HCV ruled in the possibility of unsafe injecting as a potential route of entry of HIV as well in the study community. Availability of auto-disabled syringes and needles, empowerment of the local communities and effective regulatory practices across private and public health care settings in the study area would serve as important intervention measures in such settings.

Literature search during the present investigation revealed that some of the tertiary care centers in Uttar Pradesh had recorded increasing detection of HCV among hospital attendees [23–25]. A medical institution, in the capital city of Lucknow, even highlighted that a high proportion of PLHIV (286/350; 82%) reported receiving medication through unsterile syringes and needles [26]. More than 80% of the respondents in this investigation were from rural areas.

Another probable source of HCV infection among cases and controls, all of whom were aged ≥18 year, could be contaminated blood or blood product transfusion as the units supplied from licensed blood banks in India are being screened for HCV since 2001 [27]. However, it was reassuring that no significant difference existed between cases and controls in terms of receiving blood transfusion nor with regard to invasive surgical procedures.

Our findings have close similarity to a recent iatrogenic HIV outbreak in rural Cambodia. In the Battambang province in northwest Cambodia, the family of an index HIV case, who also suffered from tuberculosis, alleged that the infection in the index case and two other family members, who became infected with HIV during the same period, were linked to medical injections received from an unlicensed health practitioner. While as high as 78% of the HIV infected individuals in this outbreak were found to be co-infected with HCV [28], in Unnao, 85% of the PLHIV had evidence of exposure to HCV infection. These two studies reveal an aspect of HCV co-infection in HIV cohorts in the Asia Pacific region, which as yet was known to range from 4% to 43% [29].

Cases ever infected with HCV, across different age groups in our study varied from 67% to 100% and in controls it ranged from 52% to 67% (Fig 2). This indicates a widespread looming risk posed to young, adults and elderly in the study community.

The arguments, presented above, raise the possibility of an iatrogenic transmission of HIV and HCV in our study setting. Discourses on the three recognized patterns of HCV epidemic [30], and evidence of fairly even presence of HCV exposure across different age groups of the study participants lend further support to this assertion. Finally, we draw attention towards the potential spread of HIV in population in other parts of Unnao as well because a rising trend of HIV was detected among ICTC attendees in the district.

We maintain that contribution of unemployment and resulting vulnerability to risk exposure during treatment seeking from settings with poor infection control could not be ruled out in our investigation. The United Nations Development Program (UNDP) ranked Unnao on 64th position among erstwhile 70 districts in the state of UP on deprivation index, which is a composite indicator of income, health and education [31]. This observation has relevance to the current investigation, as HIV is known to thrive best in the settings of poverty and under development.

The current investigation was limited by being observational in nature. We interviewed the cases and controls to record exposures to potential risk factors such as unsafe sex and injection practices through recall. Socially desirable responses around these issues and recall ability were the potential sources of bias. We therefore incorporated tests for various biomarkers in our study. Such diversity in investigation approach allowed us to triangulate the findings and draw unyielding inferences. Significantly higher proportion of cases with unsafe injection exposure during treatment compared to controls over the last five years indicated that such practices might have played a role in HIV transmission in the study setting. The monophyletic clustering of HIV sequences from cases with high homology supported that they had a common ancestry. However, the current investigation was not equipped to link this ancestry with any specific source of infection.

## 6. Conclusions

We conclude that 'experiencing unsafe injecting' in care settings was strikingly high among PLHIV compared to those who did not contract HIV. The role of such unsafe injecting cannot therefore be ruled out as a factor associated with HIV transmission in the study area. Disparately high presence of HCV Ab among HIV sero-reactive individuals compared to controls and current knowledge about HCV transmission also lend support to our assertion. We underline the need for ensuring safe injection practices in therapeutic settings in the study area at the earliest. Making auto-disabled syringes and needles available in all care settings, empowerment of the local community and effective regulatory practices will play pivotal roles in this regard.

## Supporting information

**S1 File.**
(PDF)

## Acknowledgments

Critical administrative and planning support extended from the offices of the District Magistrate of Unnao, Chief Medical Officer of Health, UP State AIDS Control Society, District Tuberculosis Officer and staff of GSVM Medical College, Kanpur is acknowledged. We also

acknowledge Dr. Tarun Roy for paying a supervisory visit during the implementation of the study and Mr. Sandeep Singh, Counselor, Mr. Hayatulla Altaf, Laboratory Technician, Ms. Pragya Singh, Field Interviewer, Mr. Moolchand Verma, Data Entry Operator for study execution. Accredited Social Health Activist (ASHA) workers, State Tuberculosis Supervisor, Coordinator—Revised National Tuberculosis Control Program and Mr. Sachin Kumar, Program Manager, Non-Government Organization and HIV targeted intervention partner- "HUMANA" took part in the study implementation plan and coordination activities. Mr. Ajit Patil, technical officer 'A'provided assistance in HIV sequencing and construction of phylogenetic tree.

Sequence data—Nucleotide sequences were submitted to GenBank under accession numbers MW016009-MW016022.

## Author Contributions

**Conceptualization:** Preety Pathak, Raman Gangakhedkar, Samiran Panda.

**Data curation:** Sandip Patil, Amrita Rao, Swarali Kurle, Arati Mane, Amit Nirmalkar, Vinita Verma.

**Formal analysis:** Sandip Patil, Amrita Rao, Swarali Kurle, Arati Mane, Amit Nirmalkar, D. C. S. Reddy, Ashwini Shete, Samiran Panda.

**Funding acquisition:** Manjula Singh, Raman Gangakhedkar.

**Investigation:** Sandip Patil, Amrita Rao, Preety Pathak, Swarali Kurle, Arati Mane, Amit Nirmalkar, A. K. Singhal, Vinita Verma, Mukesh Kumar Singh, Ashwini Shete, Manjula Singh, Raman Gangakhedkar, Samiran Panda.

**Methodology:** D. C. S. Reddy, Samiran Panda.

**Project administration:** Sandip Patil, Amrita Rao, Preety Pathak, A. K. Singhal, Vinita Verma, Mukesh Kumar Singh, D. C. S. Reddy, Manjula Singh, Raman Gangakhedkar, Samiran Panda.

**Supervision:** D. C. S. Reddy, Samiran Panda.

**Validation:** Amit Nirmalkar.

**Visualization:** Samiran Panda.

**Writing – original draft:** Sandip Patil, Amrita Rao, Swarali Kurle, Arati Mane, Samiran Panda.

**Writing – review & editing:** Preety Pathak, Swarali Kurle, Arati Mane, Amit Nirmalkar, A. K. Singhal, Vinita Verma, Mukesh Kumar Singh, D. C. S. Reddy, Ashwini Shete, Manjula Singh, Raman Gangakhedkar, Samiran Panda.

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
