## [Decision Letter · Decision Letter 0]

27 Aug 2020

PONE-D-19-32362

A case-control investigation of HIV outbreak in the northern district of Unnao, India unearths an underlying spread of HCV

PLOS ONE

Dear Dr. Panda,

Thank you for submitting your manuscript to PLOS ONE. After careful consideration, we feel that it has merit but does not fully meet PLOS ONE’s publication criteria as it currently stands. Therefore, we invite you to submit a revised version of the manuscript that addresses the points raised during the review process.

We look forward to receiving your revised manuscript.

Kind regards,

Kimberly Page, PhD, MPH

Academic Editor

PLOS ONE

Journal Requirements:

2. Please include additional information regarding the survey or questionnaire used in the study and ensure that you have provided sufficient details that others could replicate the analyses. If you developed and/or translated a questionnaire as part of this study and it is not under a copyright more restrictive than CC-BY, please include a copy, in both the original language and English, as Supporting Information.

3. PLOS ONE requires that experiments, statistics, and other analyses are performed to a high technical standard and are described in sufficient detail to allow another researcher to reproduce the experiments (https://journals.plos.org/plosone/s/criteria-for-publication#loc-3), and that the conclusions presented in the manuscript are supported by the data (https://journals.plos.org/plosone/s/criteria-for-publication#loc-4). Please therefore remove the statements throughout your manuscript that HSV-2 IgG is an objective biomarker for sexual lifestyle or exposure, since HSV-2 transmission can occur by other modes, including vertical transmission from mother to child.

4.We note that you have indicated that data from this study are available upon request. PLOS only allows data to be available upon request if there are legal or ethical restrictions on sharing data publicly. For information on unacceptable data access restrictions, please see http://journals.plos.org/plosone/s/data-availability#loc-unacceptable-data-access-restrictions.

6.We note that [Figure(s) 1] in your submission contain [map/satellite] images which may be copyrighted. All PLOS content is published under the Creative Commons Attribution License (CC BY 4.0), which means that the manuscript, images, and Supporting Information files will be freely available online, and any third party is permitted to access, download, copy, distribute, and use these materials in any way, even commercially, with proper attribution. For these reasons, we cannot publish previously copyrighted maps or satellite images created using proprietary data, such as Google software (Google Maps, Street View, and Earth). For more information, see our copyright guidelines: http://journals.plos.org/plosone/s/licenses-and-copyright.

1.    You may seek permission from the original copyright holder of Figure(s) [1] to publish the content specifically under the CC BY 4.0 license. 

Additional Editor Comments (if provided):

I agree with the two reviewers that the paper has interest and presents on an important topic. However, in its present form it does't meet criteria for publishing. Re-organization and focus will significantly improve the manuscript. In addition to editorial (grammar) review, this paper could be shortened by 20% by focusing on the main findings.

Reviewers' comments:

Reviewer's Responses to Questions

**Comments to the Author**

1. Is the manuscript technically sound, and do the data support the conclusions?

Reviewer #1: Partly

Reviewer #2: Yes

2. Has the statistical analysis been performed appropriately and rigorously? 

Reviewer #1: Yes

Reviewer #2: No

3. Have the authors made all data underlying the findings in their manuscript fully available?

Reviewer #1: No

Reviewer #2: Yes

4. Is the manuscript presented in an intelligible fashion and written in standard English?

Reviewer #1: No

Reviewer #2: Yes

5. Review Comments to the Author

Reviewer #1: Thank you for asking me to review this article under consideration. 1. The article has many typographical errors and the authors should employ the use of GRAMMARLY, (it is a free tool) to help in edits throughout the document. 2. The methodology needs to be re-written. Study area. Study design, Study population. Sample size determination. Sampling technique. Dependent and independent variables. eligibility criteria. Data collection tool and procedure; was this interviewer assisted or administered? remember to include who took the blood samples and how? Where was the tool adapted from? was it validated? any pretests? Data analysis: SPSS? Descriptive statistics computed? Ethical considerations: ethics committee and how consent was obtained from participants, how research assistants were trained on e.g confidentiality. Keeping respondents identities anonymous?

Reporting them under these sub sections, helps for clarity. many of these sub themes were either totally absent or mixed up together.

3. Discussion. Why is the first line mentioning secondary data analysis? Please clarify. The second paragraph that mentions bias should be moved to the last paragraph and how these were addressed should be included.

4. There was no conclusion or recommendation. Please kindly include this

5. An algorithm for how this cases and controls were selected should be done.

6. This artcle from pubmed can be a great guide in re-organizing the article.

https://www.ncbi.nlm.nih.gov/pmc/articles/PMC7049326/

Determinants of Anemia among HIV-Positive Children on Highly Active Antiretroviral Therapy Attending Hospitals of North Wollo Zone, Amhara Region, Ethiopia, 2019: A Case-Control Study

7. The conclusion and recommendations should please be in response to the objectives of the paper, and the results respectively.

Thank you for the opportunity

Reviewer #2: This is an interesting study on a major public health issue observed in several Asian countries where the use of unsterilized injection equipment for medical use and outbreaks of HCV or HIV are common. However, there are several issues in the current manuscripts:

1. The population is not adequately described ,especially for an international reader. A small table including the total population of Uttar Pradesh, Unnao district, Bangarmau block and Premganj township-villages of Karimuddinpur and Chakmeerapur should be included.

2. The history of blood transfusion and invasive surgical procedures is not included in the tables.

3. The statistics in the multivariate analysis are not adequate. I recommend to present models including "Received intramuscular injection in last five years" or "Received intravenous fluid for treatment in last five years" in addition to the current model including "Syringe & needle used during intramuscular injection in last five years". The last model may be problematic since the potencial for recall bias is high.

4. In figure 3 the legend does not describe the meaning of different colours.

5. I recommend modification of the title to emphasise the main findings of the study e.g. unsterilized injection equipment for medical use is a cause of the HIV outbreak and extremely high prevalence of HCV.

6. The authors should discuss the generalisability of the HCV findings for Unnao district and Uttar Pradesh. Their discussion is very confusing in this very important issue.

6. PLOS authors have the option to publish the peer review history of their article (what does this mean?). If published, this will include your full peer review and any attached files.

Reviewer #1: No

Reviewer #2: **Yes: **Angelos Hatzakis MD PHD

---

## [Author Response · Author response to Decision Letter 0]

29 Oct 2020

“RESPONSE TO EDITOR AND REVIEWERS”

EDITOR’S COMMENTS

We have ensured that manuscript meets PLOS ONE's style requirements, including those for file naming.

2. Please include additional information regarding the survey or questionnaire used in the study and ensure that you have provided sufficient details that others could replicate the analyses. If you developed and/or translated a questionnaire as part of this study and it is not under a copyright more restrictive than CC-BY, please include a copy, in both the original language and English, as Supporting Information.

We have provided a copy of questionnaire developed by us for the study.

3. PLOS ONE requires that experiments, statistics, and other analyses are performed to a high technical standard and are described in sufficient detail to allow another researcher to reproduce the experiments (https://journals.plos.org/plosone/s/criteria-for-publication#loc-3), and that the conclusions presented in the manuscript are supported by the data (https://journals.plos.org/plosone/s/criteria-for-publication#loc-4). Please therefore remove the statements throughout your manuscript that HSV-2 IgG is an objective biomarker for sexual lifestyle or exposure, since HSV-2 transmission can occur by other modes, including vertical transmission from mother to child.

We have removed the statements throughout the manuscript- “HSV-2 IgG is an objective biomarker for sexual lifestyle or exposure”

 4.We note that you have indicated that data from this study are available upon request. PLOS only allows data to be available upon request if there are legal or ethical restrictions on sharing data publicly. For information on unacceptable data access restrictions, please see http://journals.plos.org/plosone/s/data-availability#loc-unacceptable-data-access-restrictions.

Data contain potentially identifying or sensitive patient information / study location specific information which, if disclosed, may lead to stigma and discrimination. Due to ethical restrictions, data can be made available on request to Institutional Ethics committee of ICMR-NARI, Pune. (Email id- ecnari@nariindia.org) as well as Project Director,Uttar Pradesh State AIDS Control Society (UPSACS) (Email id- pd.upsacs@gmail.com)

 ORCID ID has been updated

6.We note that [Figure(s) 1] in your submission contain [map/satellite] images which may be copyrighted. All PLOS content is published under the Creative Commons Attribution License (CC BY 4.0), which means that the manuscript, images, and Supporting Information files will be freely available online, and any third party is permitted to access, download, copy, distribute, and use these materials in any way, even commercially, with proper attribution. For these reasons, we cannot publish previously copyrighted maps or satellite images created using proprietary data, such as Google software (Google Maps, Street View, and Earth). For more information, see our copyright guidelines: http://journals.plos.org/plosone/s/licenses-and-copyright.

We have excluded map (Fig 1 from earlier version) from revised manuscript and content of the map has been tabulated.

Additional Editor Comments:

I agree with the two reviewers that the paper has interest and presents on an important topic. However, in its present form it does't meet criteria for publishing. Re-organization and focus will significantly improve the manuscript. In addition to editorial (grammar) review, this paper could be shortened by 20% by focusing on the main findings.

We have used Grammarly tool for grammar review. We have also reorganized and reduced our content by focusing on main findings to improve manuscript.

Reviewer 1 

1. The article has many typographical errors and the authors should employ the use of GRAMMARLY, (it is a free tool) to help in edits throughout the document. 

We have used Grammarly tool which was immensely helpful for grammar review.

2. The methodology needs to be re-written. Study area. Study design, Study population. Sample size determination. Sampling technique. Dependent and independent variables. eligibility criteria. Data collection tool and procedure; was this interviewer assisted or administered? remember to include who took the blood samples and how? Where was the tool adapted from? was it validated? any pretests? Data analysis: SPSS? Descriptive statistics computed? Ethical considerations: ethics committee and how consent was obtained from participants, how research assistants were trained on e.g confidentiality. Keeping respondents identities anonymous?

Reporting them under these sub sections, helps for clarity. many of these sub themes were either totally absent or mixed up together.

We have reorganized our manuscripts and incorporated all aforementioned suggestions in methodology section.

3. Discussion. Why is the first line mentioning secondary data analysis? Please clarify. 

We shifted it to last part of the first paragraph of discussion.

4. The second paragraph that mentions bias should be moved to the last paragraph and how these were addressed should be included.

We have moved bias part to the last paragraph

5. There was no conclusion or recommendation. Please kindly include this

We have included conclusion section in the manuscript.

6. An algorithm for how this cases and controls were selected should be done.

We have added an algorithm (Fig 1) pertaining to selection of cases and controls.

7. The conclusion and recommendations should please be in response to the objectives of the paper, and the results respectively.

Conclusion in response to the objectives of the paper and results have been included.

8. This article from pubmed can be a great guide in re-organizing the article.

https://www.ncbi.nlm.nih.gov/pmc/articles/PMC7049326/

Determinants of Anemia among HIV-Positive Children on Highly Active Antiretroviral Therapy Attending Hospitals of North Wollo Zone, Amhara Region, Ethiopia, 2019: A Case-Control Study

We have followed the structure as depicted in aforementioned article suggested by reviewer. 

Reviewer 2

1. The population is not adequately described ,especially for an international reader. A small table including the total population of Uttar Pradesh, Unnao district, Bangarmau block and Premganj township-villages of Karimuddinpur and Chakmeerapur should be included.

 We have adequately described population in revised manuscript with reference to 

 Census 2011.

2. The history of blood transfusion and invasive surgical procedures is not included in the tables.

 We have added aforementioned history in table 4.

3. The statistics in the multivariate analysis are not adequate. I recommend to present models including "Received intramuscular injection in last five years" or "Received intravenous fluid for treatment in last five years" in addition to the current model including "Syringe & needle used during intramuscular injection in last five years". The last model may be problematic since the potencial for recall bias is high.

We added “receipt of intramuscular injection” and rerun multivariate regression analysis and results have been produced in table 5.

4. In figure 3 the legend does not describe the meaning of different colours.

We have excluded fig 3 and new phylogenetic tree has been added. We also modified methodology pertaining to phylogenetic analysis

5. I recommend modification of the title to emphasise the main findings of the study e.g. unsterilized injection equipment for medical use is a cause of the HIV outbreak and extremely high prevalence of HCV.

 As per suggestion, we have modified the title of the manuscript.

6. The authors should discuss the generalisability of the HCV findings for Unnao district and Uttar Pradesh. Their discussion is very confusing in this very important issue.

 We have discussed generalizability of the HCV findings for Unnao district and Uttar 

 Pradesh. We have simplified the discussion and made it more lucid.

---

## [Decision Letter · Decision Letter 1]

24 Nov 2020

Unsterile injection equipment associated with HIV outbreak and an extremely high prevalence of HCV - a case-control investigation from Unnao, India

PONE-D-19-32362R1

Dear Dr. Panda,

We’re pleased to inform you that your manuscript has been judged scientifically suitable for publication and will be formally accepted for publication once it meets all outstanding technical requirements.

Kind regards,

Kimberly Page, PhD, MPH

Academic Editor

PLOS ONE

Additional Editor Comments (optional):

Reviewers' comments:

Reviewer's Responses to Questions

**Comments to the Author**

1. If the authors have adequately addressed your comments raised in a previous round of review and you feel that this manuscript is now acceptable for publication, you may indicate that here to bypass the “Comments to the Author” section, enter your conflict of interest statement in the “Confidential to Editor” section, and submit your "Accept" recommendation.

Reviewer #1: All comments have been addressed

Reviewer #2: All comments have been addressed

2. Is the manuscript technically sound, and do the data support the conclusions?

Reviewer #1: (No Response)

Reviewer #2: Yes

3. Has the statistical analysis been performed appropriately and rigorously? 

Reviewer #1: (No Response)

Reviewer #2: Yes

4. Have the authors made all data underlying the findings in their manuscript fully available?

Reviewer #1: (No Response)

Reviewer #2: Yes

5. Is the manuscript presented in an intelligible fashion and written in standard English?

Reviewer #1: (No Response)

Reviewer #2: Yes

6. Review Comments to the Author

Reviewer #1: (No Response)

Reviewer #2: (No Response)

7. PLOS authors have the option to publish the peer review history of their article (what does this mean?). If published, this will include your full peer review and any attached files.

Reviewer #1: No

Reviewer #2: **Yes: **Angelos Hatzakis

---

## [Editor Report · Acceptance letter]

27 Nov 2020

PONE-D-19-32362R1 

Unsterile injection equipment associated with HIV outbreak and an extremely high prevalence of HCV - a case-control investigation from Unnao, India 

Dear Dr. Panda:

I'm pleased to inform you that your manuscript has been deemed suitable for publication in PLOS ONE. Congratulations! Your manuscript is now with our production department. 

Kind regards, 

on behalf of

Dr. Kimberly Page 

Academic Editor

PLOS ONE